# Genomic Legacies of Ancient Adaptation Illuminate GC-Content Evolution in Bacteria

Wenkai Teng,[a] Bin Liao,[a] Mengyun Chen,[b] Wensheng Shu[b]

aSchool of Life Sciences, Sun Yat-sen University, Guangzhou, Guangdong, China
bSchool of Life Sciences, South China Normal University, Guangzhou, Guangdong, China

**ABSTRACT** Bacterial evolution is characterized by strong purifying selection as well as rapid adaptive evolution in changing environments. In this context, the genomic GC content (genomic GC) varies greatly but presents some level of phylogenetic stability, making it challenging to explain based on current hypotheses. To illuminate the evolutionary mechanisms of the genomic GC, we analyzed the base composition and functional inventory of 11,083 representative genomes. A phylogenetically constrained bimodal distribution of the genomic GC, which mainly originated from parallel divergences in the early evolution, was demonstrated. Such variation of the genomic GC can be well explained by DNA replication and repair (DRR), in which multiple pathways correlate with the genomic GC. Furthermore, the biased conservation of various stress-related genes, especially the DRR-related ones, implies distinct adaptive processes in the ancestral lineages of high- or low-GC clades which are likely induced by major environmental changes. Our findings support that the mutational biases resulting from these legacies of ancient adaptation have changed the course of adaptive evolution and generated great variation in the genomic GC. This highlights the importance of indirect effects of natural selection, which indicates a new model for bacterial evolution.

**IMPORTANCE** GC content has been shown to be an important factor in microbial ecology and evolution, and the genomic GC of bacteria can be characterized by great intergenomic heterogeneity, high intragenomic homogeneity, and strong phylogenetic inertia, as well as being associated with the environment. Current hypotheses concerning direct selection or mutational biases cannot well explain these features simultaneously. Our findings of the genomic GC showing that ancient adaptations have transformed the DRR system and that the resulting mutational biases further contributed to a bimodal distribution of it offer a more reasonable scenario for the mechanism. This would imply that, when thinking about the evolution of life, diverse processes of adaptation exist, and combined effects of natural selection should be considered.

**KEYWORDS** bacteria, GC content, DNA repair, adaptation, natural selection, mutational biases, eubacteria

Bacteria are usually supposed to be subject to a high level of purifying selection in light of huge effective population sizes (1). The situation hence can become confusing when it comes to how bacteria have generated great variation in their genomic base composition (i.e., genomic GC content [GC content]) during their adaptive evolution (2). The genomic GC content of bacteria, which can vary enormously from below 20% to nearly 75%, is not normally distributed at odds with the expectation from an entirely stochastic model (3, 4). Therefore, the variation in genomic GC has been related to natural selection acting directly on DNA or protein sequences in previous studies (5, 6). When bacteria are exposed to environmental stresses such as oxidative stress, heat stress, and nutrient limitation (e.g., nitrogen or carbon limitation), the chemical property differences of

Address correspondence to Wensheng Shu, shuwensheng@m.scnu.edu.cn, or Mengyun Chen, chenmy33@m.scnu.edu.cn.

The authors declare no conflict of interest.

base pairs or amino acids likely influence the genomic GC, provided that the fitness differences are large enough (7–9). The correlations detected between the genomic GC and a few selective agents favor an environmental constraint on the base composition to some extent, though often insufficiently (10–14).

In conventional views, bacteria evolve rapidly in response to environmental changes, leaving most genetic variations nonpersistent (15). However, previous studies have revealed remarkable phylogenetic inertia in the genomic GC, hinting at a certain level of stability of the base composition despite ever-changing environments over long timescales (13, 16). Thus, some attention has shifted from direct selection toward the effect of genetically conserved components, the DNA replication and repair (DRR) proteins, which potentially cause mutational biases and introduce intricate effects on the genomic GC (17, 18).

Bacteria suffer various DNA damages at any moment during growth, especially under environmental stresses (19). For example, the cytosine in DNA can be spontaneously deaminated into uracil, and the guanine is susceptible to oxidative damage (20, 21). These lesions could give rise to different types of mutations in DNA replication (DR) and directly influence the base composition (22). Nevertheless, multiple pathways can be recruited to cope with the DNA damages, including base excision repair (BER), nucleotide excision repair (NER), nonhomologous end joining (NHEJ), mismatch repair (MMR), and homologous recombination (HR) (23). A particularly striking case is BER, in which diverse lesions including the mutagenic uracil and oxidized guanine in DNA are recognized and then removed by DNA glycosylases (22, 24). The proteins involved in these pathways are critical for the response to environmental stresses, and deletions of them could contribute to an elevated rate of related mutations (25). On the other hand, the error-prone polymerases involved in translesion synthesis (TLS) can trigger various replication errors in the newly synthesized DNA strand, thus affecting the mutagenic spectrum (26–28). To date, several DRR-related proteins have been considered responsible for the variation in genomic GC (18, 29–31). However, it remains enigmatic how the DRR system has evolved during bacterial adaptation to drive the genomic GC, which also correlates with the environment (13).

As an essential genome feature, GC content can be related to many aspects of the genome, especially amino acid and codon usage due to high coding density, and therefore, a better understanding of it will profoundly promote the comprehension of bacterial evolution (32, 33). In order to clarify the evolutionary mechanisms of genomic GC, we analyzed all currently available representative genomes of bacteria. The distribution pattern and phylogenetic history of genomic GC and correlated genome functions were investigated. Our results suggest an indirect method of natural selection in which the ancient adaptations have transformed the bacterial genome (especially the DRR system), contributing to a bimodal distribution pattern of genomic GC.

## RESULTS

**Early evolution contributes to a bimodal distribution of genomic GC.** Among 11,083 high-quality bacterial representative genomes (see Fig. S1 and Table S1 in the supplemental material), the genomic GC content varies greatly from about 16% to 77% and, surprisingly, displays a bimodal distribution pattern (Fig. 1A). This indicates a remarkable divergence of the genomic GC, as most genomes have GC content either below 45% or above 60% (Fig. 1A). Scanning the genomic GC across different taxonomic levels shows a high level of phylogenetic inertia, and particularly, more than 60% of the total variance can be explained at the phylum level (Fig. 1B). Consistently, such bimodal distribution with 50% in the middle nearly disappears at the phylum level (Fig. S2). Distributions of genomic GC in most phyla are approximatively unimodal, with peak values either greater or less than 50% (which can be defined as high- or low-GC clade, respectively), especially in those having plenty of representatives, i.e., *Proteobacteria*, *Firmicutes*, *Actinobacteria*, and *Bacteroidetes* (Fig. 1C and D). A strong phylogenetic signal can also be reflected by the high values of Blomberg's $K$ ($K = 1.47$) and Pagel's $\lambda$ ($\lambda = 0.998$) in the bacterial tree (Fig. 2). The estimated kappa ($\kappa = 0.842$) and delta ($\delta = 0.567$) on genomic GC further suggest a punctuated and decelerated

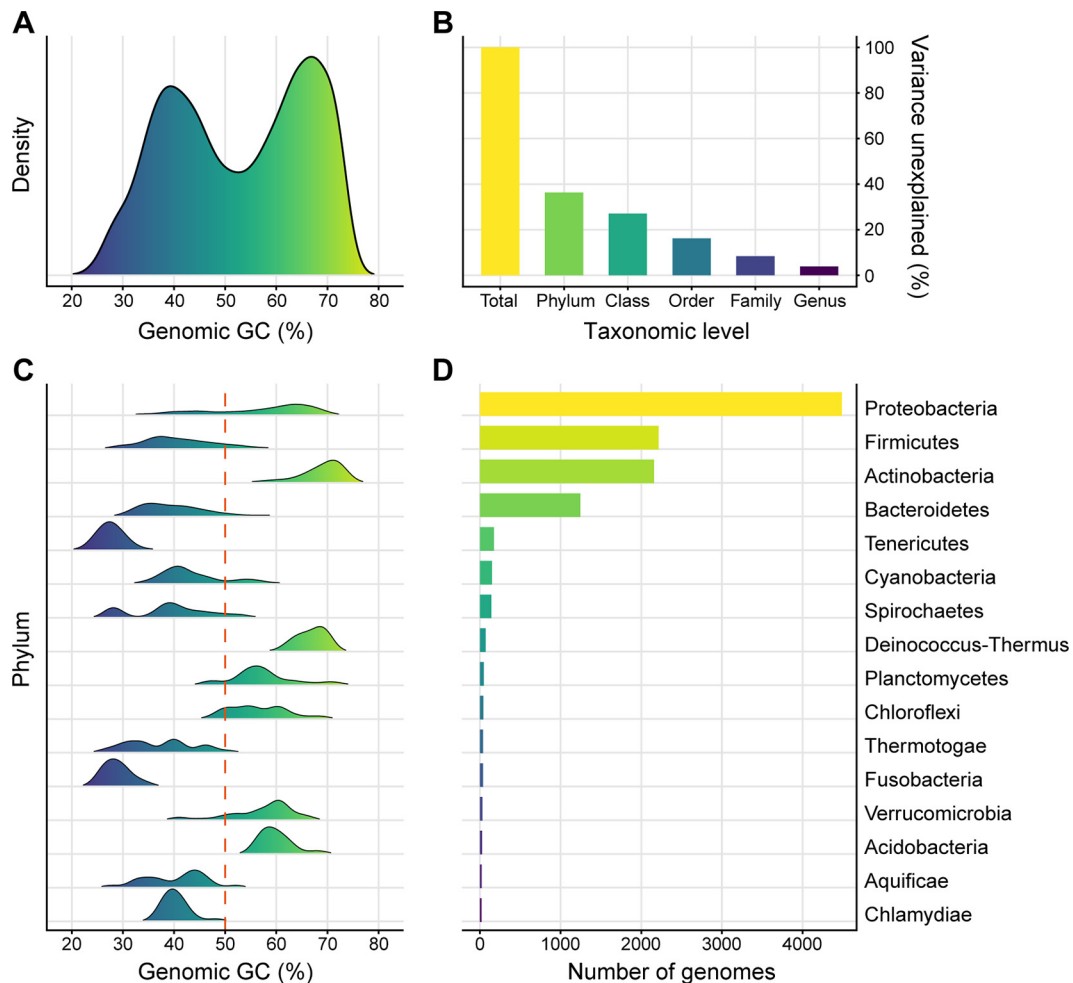

**FIG 1** Distribution pattern and phylogenetic inertia of genomic GC content. (A) Distribution of the genomic GC of 11,083 bacterial representative genomes. (B) Variance of the genomic GC unexplained by bacterial taxonomy at different levels. (C) Distribution of the genomic GC of the phyla with more than 20 representative genomes. (D) Number of representative genomes of phyla shown in panel C.

evolution mode (Fig. 2, see Materials and Methods). The results described above suggest that the early evolution of bacteria contributes much to the divergence of the genomic GC, which leads to the bimodal distribution.

Ancestral state reconstruction further reveals multiple shifts in the genomic GC across the evolution of bacteria. Low-GC clades are interspersed among high-GC clades on the phylogenetic tree even within the same bacterial group (e.g., the Terrabacteria group, the *Proteobacteria* group, and the FCB and PVC group classified by the NCBI taxonomy; https://www.ncbi.nlm.nih.gov/Taxonomy/) (Fig. 2). In detail, while *Firmicutes*, *Tenericutes*, and *Cyanobacteria* have relatively low genomic GC in the Terrabacteria group, clades including *Actinobacteria*, *Deinococcus-Thermus*, and *Chloroflexi* possess genomes with comparatively high GC content (Fig. 1C and 2). When it comes to *Proteobacteria*, the genomic GC of most lineages, especially those from *Alphaproteobacteria*, *Betaproteobacteria*, and *Deltaproteobacteria*, is rather high (Fig. 2). Despite that, low-GC clades are embedded inside this group as well, taking the *Epsilonproteobacteria* clade as an example (Fig. 2, Fig. S3). In particular, in the *Gammaproteobacteria* clade, with a huge phylogenetic diversity, the genomic GC has diverged multiple times and displays an analogous bimodal distribution (Fig. 2, Fig. S3). Furthermore, the situation in the clade containing the FCB and PVC group is also similar, where *Bacteroidetes* and chlamydiae have relatively low values of genomic GC, in contrast to their relatives *Planctomycetes* and *Verrucomicrobia* (Fig. 1C and 2). Collectively, these results suggest a

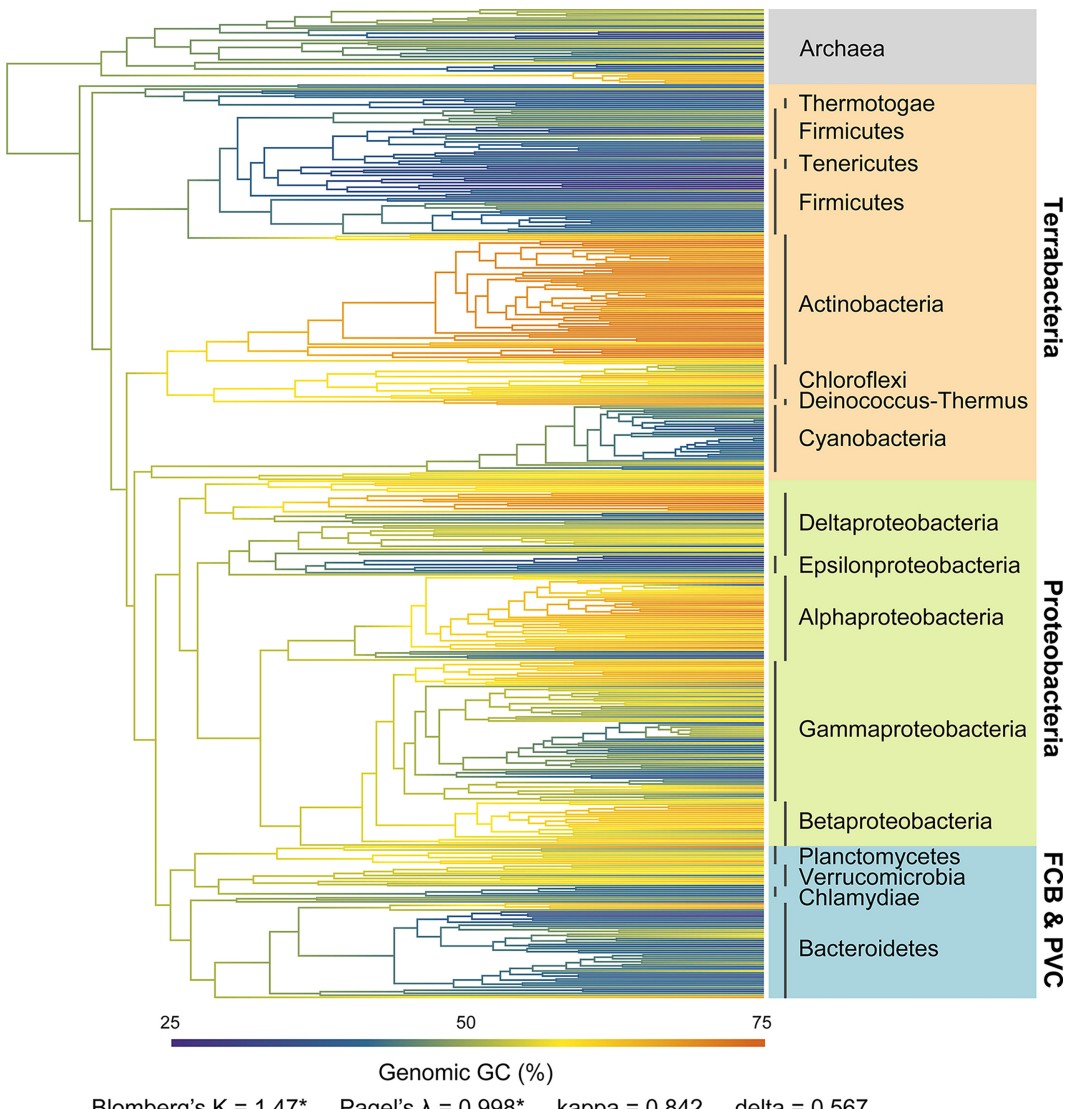

Blomberg's K = 1.47*     Pagel's λ = 0.998*     kappa = 0.842     delta = 0.567

**FIG 2** Ancestral state reconstruction of genomic GC content. A phylogenetic tree of the representatives of 450 bacterial families with archaeal genomes as the outgroup was used in the analysis. Divergence times were estimated using the R package ape. The genomic GC contents of ancestral nodes were reconstructed based on the average genomic GC of each family. Phylogenetic signal estimates, Blomberg's $K$, Pagel's $\lambda$, $\kappa$, and $\delta$, of the genomic GC data are labeled below the tree. Asterisks indicate a $P$ value of $<0.01$.

likely parallelism of the GC-content divergence among bacterial genomes during the early evolution of phyla or classes and in the subsequent evolution of certain clades.

**DNA replication and repair explain the major variation in genomic GC.** Some previous studies suggested direct selection upon the usage of amino acids and synonymous codons which potentially influence the genomic GC (8, 34). Indeed, strong correlations between them were also recovered in this study (Fig. S4). However, there are several findings challenging this view. First, the correlation between the genomic GC and the usage of an amino acid depends on the average GC content of its codons rather than the chemical properties generally targeted by selection (Fig. S4). Second, the GC content genome wide ($GC$), the GC content of coding sequences ($GC_{CDS}$), the GC content of noncoding sequences ($GC_{NCS}$), and the GC content contributed by the usage of amino acids ($GC_{AA}$) and codons ($GC_{Codon}$) are all highly correlated with each other (Fig. S5). Within this context, assuming an environmental factor which equivalently influences the coding and noncoding sequences via direct selection can be confusing (5). Finally, the phylogenetic inertia of the genomic GC as shown above highlights the view of other studies that the DRR system

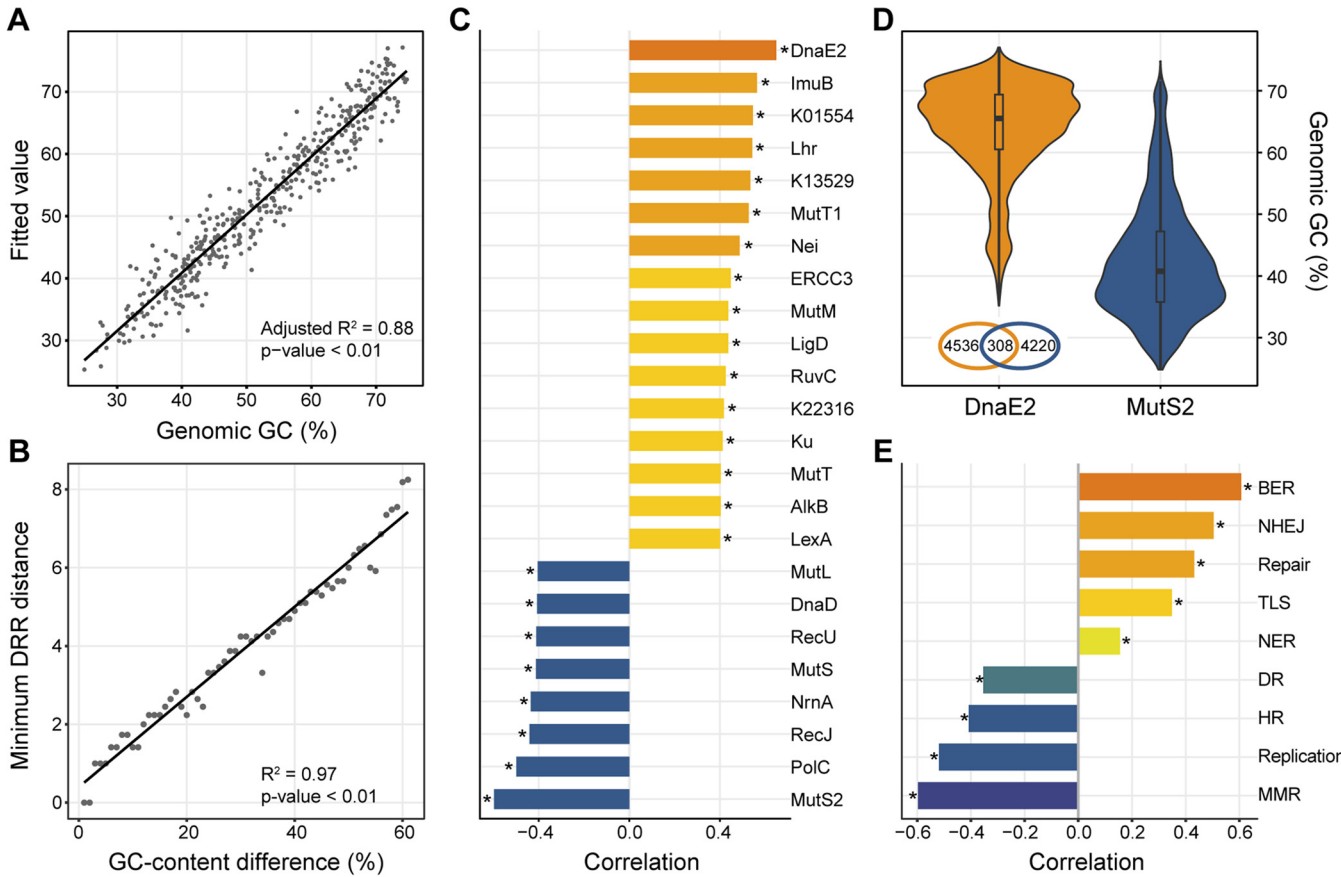

**FIG 3** Relationships between genomic GC and DNA replication and repair (DRR) system. (A) Multiple regression analysis of the genomic GC and 217 DRR-related KOs (presenting in more than 3 families) based on the phylogenetic generalized least-squares (PGLS) analysis. Fitted values of the genomic GC content are shown in contrast to real values. (B) Minimum Euclidean distance in the DRR system with changing differences in the GC content among all genome pairs. (C) Correlations between the genomic GC and DRR-related KOs. Of all 231 KOs, only the one with a considerable (Pearson's $r > 0.4$ or $r < -0.4$) correlation is shown. (D) Comparison of the GC contents of genomes containing the DnaE2 and/or MutS2. The Venn diagram inside the graph shows the number of genomes containing DnaE2 and/or MutS2. (E) Correlations between the genomic GC and DRR-related KEGG pathways or BRITE categories (DR, DNA replication; BER, base excision repair; NER, nucleotide excision repair; MMR, mismatch repair; HR, homologous recombination; NHEJ, nonhomologous end-joining; TLS, translesion synthesis; replication, DNA replication proteins; repair, DNA repair and recombination proteins). The asterisks in panels C and E indicate an adjusted $P$ value of <0.01.

mainly drives the variation in genomic GC and synchronously influences the amino acid and codon usage (18, 31, 35).

As anticipated, a linear model using 217 DRR-related KEGG orthologs (KOs) can explain up to 88% of the total variance (that is, with a multiple correlation coefficient of 0.94), even taking phylogenetic relationships into account, suggesting that the DRR system can serve as a good predictor of the genomic GC (Fig. 3A). In order to test whether a change in the DRR system is necessary to alter the genomic GC, we analyzed the GC-content difference and DRR distance (a Euclidean distance calculated based on the presence of DRR-related KOs) among all genome pairs. Good agreement was achieved between the GC-content difference and allowed minimum DRR distance (the minimum distance in DRR of genome pairs within a 1% range of GC-content difference, e.g., 0 to 1%, 1 to 2%...), providing further support for our hypothesis (Fig. 3B). Correlation coefficients were also quantified for detecting the relationships between GC content and 231 DRR-related KOs detected in total using 11,083 genomes. Unlike previous scenarios (18, 30, 31), we find that multiple DRR-related KOs are implicated in the explanation of the variation of the genomic GC (Fig. 3C). Among these, DnaE2, an error-prone TLS polymerase, and MutS2, a homologue of MMR protein MutS, have the highest positive and negative correlation, respectively (Fig. 3C). The overwhelming majority of bacterial genomes selectively keep DnaE2 or MutS2 roughly according to their genomic GC (Fig. 3D).

Pathway analysis indicates that the positively correlated proteins are mainly involved in BER (MutM, Nei, and K13529), NHEJ (Ku and LigD), TLS (DnaE2 and ImuB), NER (ERCC3), and sanitization of the nucleotide pool (MutT, MutT1, and K01554) (Fig. 3C). Many of these proteins have been shown to be crucial for growth under environmental stresses. For instance, both the MutM and Ku play important roles in combating heat as well as oxidative stresses (25, 30). Beyond that, MutT and its analogue MutT1 participate in the detoxification of oxidized guanine nucleotides (21, 36). TLS polymerases are also important for growth under harsh conditions such as starvation and high temperatures (26). In stark contrast, some of the MMR- and HR-related proteins, including MutS, MutL, RecJ, RecU, and the aforementioned MutS2, are negatively correlated with genomic GC (Fig. 3C). Overall, the DRR system displays a distinct variation pattern of positively correlated BER, TLS, NHEJ, and NER versus negatively correlated DR, HR, and MMR (Fig. 3E). This variation pattern could also be observed when excluding the phylogenetic influence by performing phylogenetically independent contrast, with one exception (the insignificant negatively correlated NER) (Fig. S6). In detail, consistent with the inferred parallel divergences of genomic GC, a similar evolutionary pattern of the DRR exists in all three major clades (Fig. 2, Fig. S7).

**Other genomic legacies correlated with the divergence of the genomic GC.** In order to better understand the evolutionary scenarios, a genome-wide investigation into functional variation was conducted using all annotated KOs (9,924 in total). The results demonstrate that plentiful proteins with various functions correlate with genomic GC, with coefficients even larger than the DRR-related DnaE2 (Fig. 4A). Similar to the situation in the DRR system, the highly correlated proteins can be related to the bacterial response to environmental changes of multiple factors, including oxygen, temperature, nutrition, and osmotic pressure (Fig. 4A, Table S2). For instance, YbbN, a molecular chaperone of the $\beta$-clamp in the replication complex, can promote bacterial acclimation to higher temperatures and is mainly conserved in high-GC clades (Fig. S8). Another positively correlated protein, OtsA, is involved in the synthesis of trehalose in response to heat, cold, and osmotic stresses (Fig. 4A, Table S2). In addition, FNR, SURF1, and YgfZ, which play important roles in aerobic growth, also positively correlate with the genomic GC. In contrast, several proteins, including PepT, ComEB, and PfkA, which are required for the utilization of diverse nutrient sources, and the anaerobic regulatory proteins FLP and Rny, negatively correlate with the genomic GC. These correlations reveal a much closer relationship between the evolution of the genomic GC and that of the genome function, not just the DRR system (Fig. 4B).

Further analysis indicates that parallel gene loss likely contributes substantially to the biased conservations of functional proteins. First, most highly correlated proteins can be observed strongly conserved in distantly related lineages (Fig. S9 and S10). For example, positively correlated KOs, including DnaE2, ImuB, MutM, MutT, FNR, and YbbN, are highly conserved in *Actinobacteria*, which belong to the Terrabacteria group, as well as in *Alpha-* and *Betaproteobacteria* (Fig. S9 and S10). Thus, the related genes should have been present in the last common ancestor (LCA) of the two groups and lost in parallel during the later evolution. Further, some of the highly correlated KOs such as MutT and Lhr are even quite conserved in archaea, suggesting much earlier origins of their genes (Fig. S11). Second, there are many more KOs positively correlated with the genomic GC, which implies more severe loss of genes in low-GC clades (Fig. S12). Such loss of positively correlated genes is particularly striking for the low-GC lineages in *Gammaproteobacteria*, where the divergence of the genomic GC appeared much more recently (Fig. 4C and D). Consistent with this, the genome size of bacteria positively correlates with the genomic GC (Fig. S13). Notably, the highly correlated KOs are either quite conserved or almost entirely lost in a certain lineage, indicating that they are genomic legacies from early evolution (Fig. S14).

**Bacterial evolution and major environmental changes in early history.** According to the molecular dating of the bacterial tree, multiple important divergence events of the genomic GC content likely occurred during the period from the mid-Archaean to the Paleoproterozoic (i.e., early stages of the evolution of major clades), accompanied by

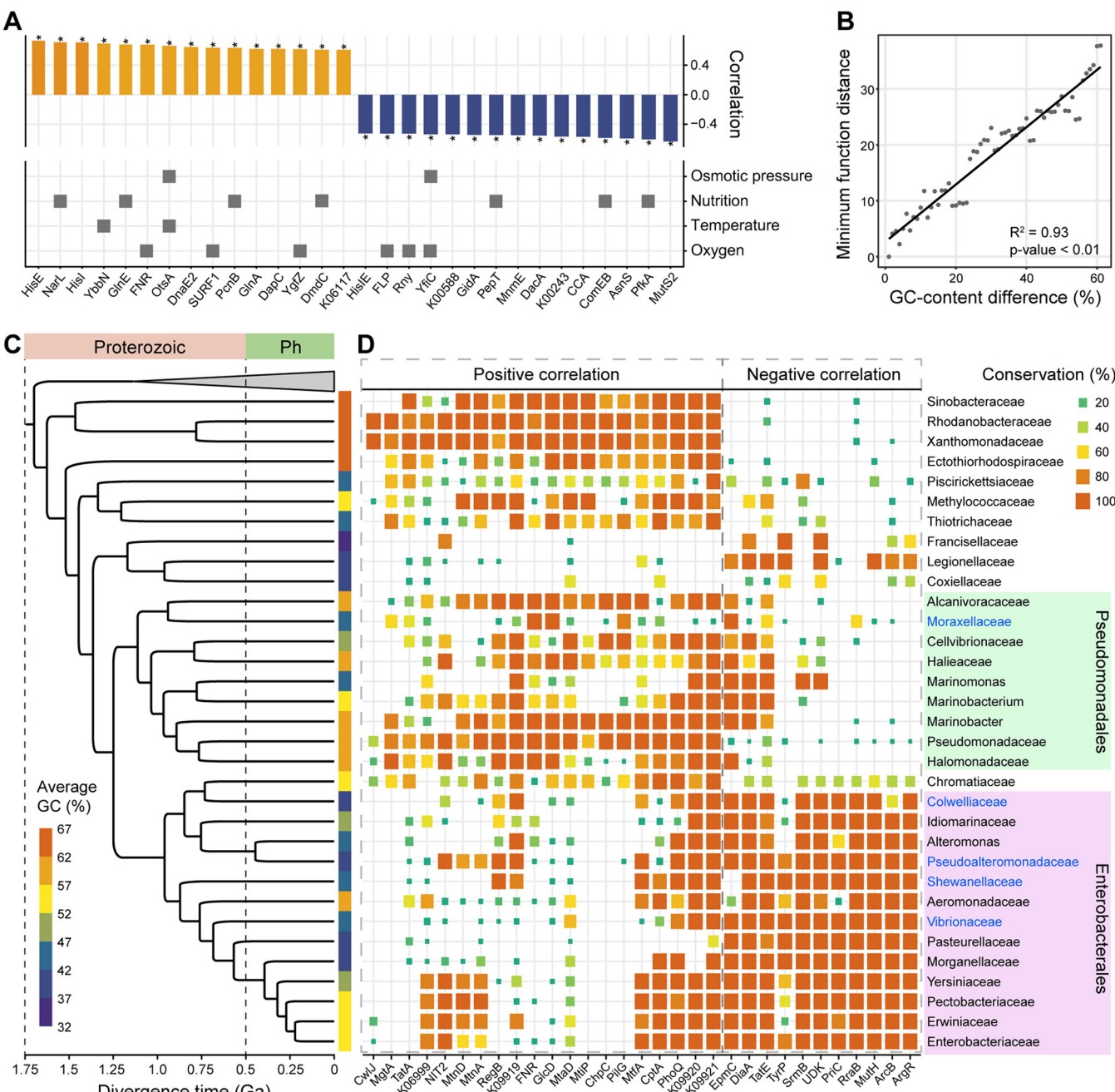

**FIG 4** Relationships between genomic GC content and genome function. (A) Highly correlated KOs and their involvement in the bacterial response to changes in four environmental factors. Of all 9,924 KOs, only the one with a considerable (Pearson's $r > 0.6$ or $r < -0.5$) correlation are shown. The asterisks beside the bars indicate an adjusted $P$ value of <0.01. (B) Minimum Euclidean distance in genome function (calculated using 9,924 KOs) with changing differences in the GC content among all genome pairs. (C) Phylogenetic tree of the major lineages in *Gammaproteobacteria* with estimated divergence times (Ga, billion years ago). Genomes belonging to the *Betaproteobacteria* were used as the outgroup. Color strips beside the tree reflect the average genomic GC content. (D) Conservation of KOs within each major lineage belonging to the *Gammaproteobacteria*. The names of lineages with the presence of more than 5 cold-adapted species are shown in blue. Only KOs with a considerable (Pearson's $r > 0.5$ or $r < -0.45$) correlation are shown. Positively correlated KOs and negatively correlated KOs are indicated using dashed boxes.

the loss of many conserved genes (Fig. 2 and Fig. 5). As revealed in a previous study, there was a peak genetic innovation of bacterial ancestors, named the Archaean genetic expansion, and rapid gene loss followed it in different lineages (37). This is in line with our findings (Fig. 5). Bacterial evolution through gene loss has commonly been associated with the adaptation to new environments (38–40). Biased conservations of the functional genes between high- and low-GC clades suggest adaptation to different environmental conditions

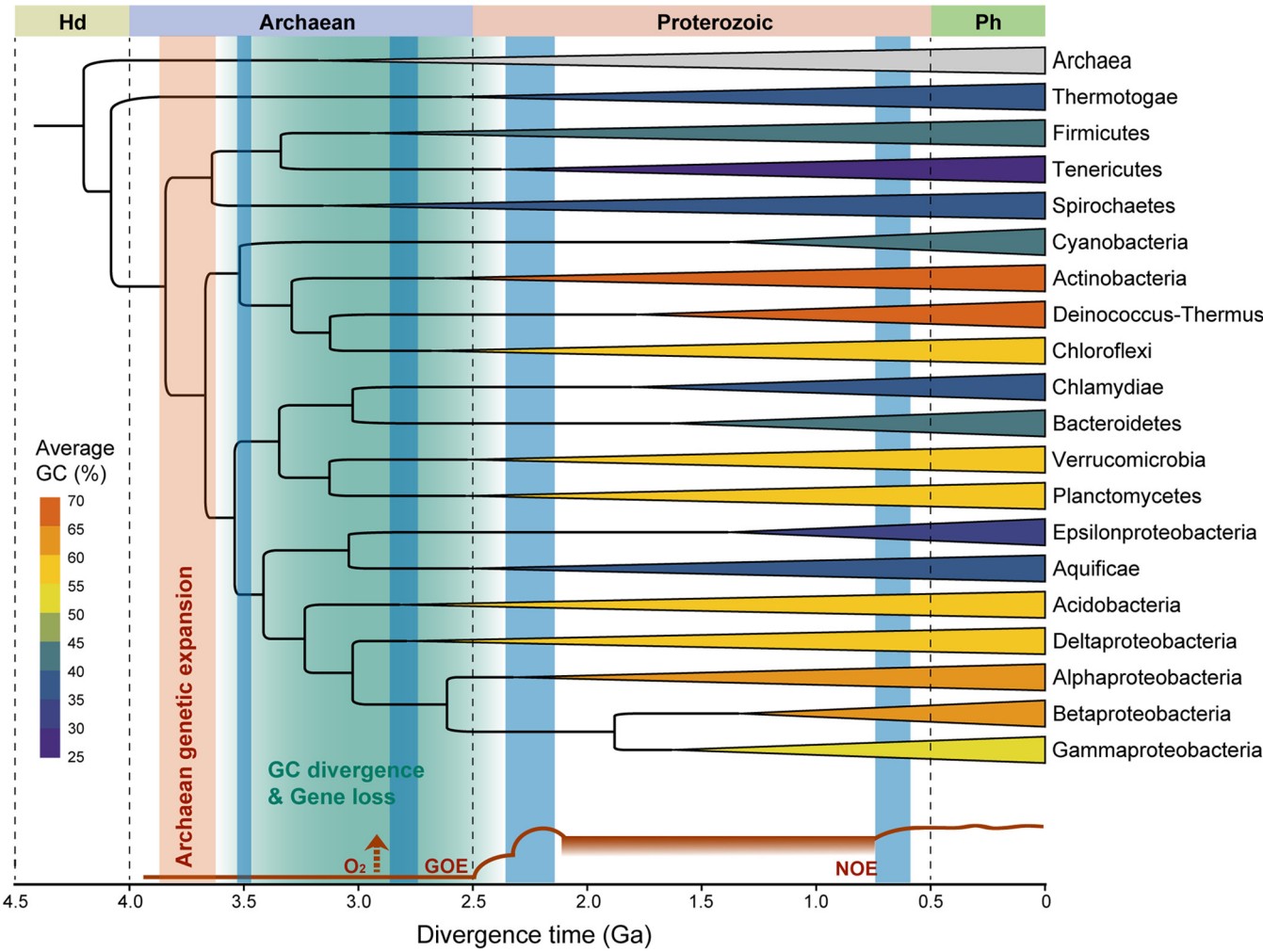

**FIG 5** Molecular clock analysis of major bacterial clades. Divergence times were estimated using the MCMC algorithm (Ga, billion years ago). Archaeal genomes were used as the outgroup. Clades are colored according to their average genomic GC. The green shaded area indicates the time period when the divergence of the genomic GC and gene loss might have occurred. The dark red bar indicates the inferred Archaean genetic expansion (37). The dark red curve at the bottom of the plot indicates the values of atmospheric oxygen throughout Earth's history (GOE, Great Oxidation Event; NOE, Neoproterozoic Oxygenation Event) (41). Time periods of glaciation before the Phanerozoic are denoted by blue bars (45).

mainly arising from differential heat and oxidative stress (Fig. 3C and 4A). Further, the evolutionary pattern of the DRR system as demonstrated above also implies different mechanisms of adaptation in high- or low-GC bacteria from different groups (Fig. 3E, Fig. S7).

The Earth's history is truly characterized by dramatic changes in atmospheric oxygen and temperature (41–43), represented by two oxidation events and a series of glaciation events (Fig. 5). Although the timing of the first emergence of oxygen in the Earth remains highly uncertain (Fig. 5), molecular analysis has revealed that all major bacterial clades except for the *Thermotogae* descend from a universal oxygen ancestor before the Great Oxidation Event (44). Based on the loss of various stress-related genes and the conservation of PepT, ComEB, and PfkA, ancestral lineages of low-GC bacteria likely have experienced strong selection by oligotrophic environments with low temperature and oxygen. This is consistent with the current knowledge about Archaean climate (Fig. 5); though it was mostly temperate to hot, the earliest evidence for glaciation dates back to the early Archaean (45).

To further investigate this relationship, we collected cold-adapted species, including the defined psychrophilic and psychrotolerant bacteria, which have been isolated from similar environments such as ocean depths, polar regions, and subglacial sediments (Table S3). As anticipated, both the psychrophilic and psychrotolerant bacteria have a skewed bimodal

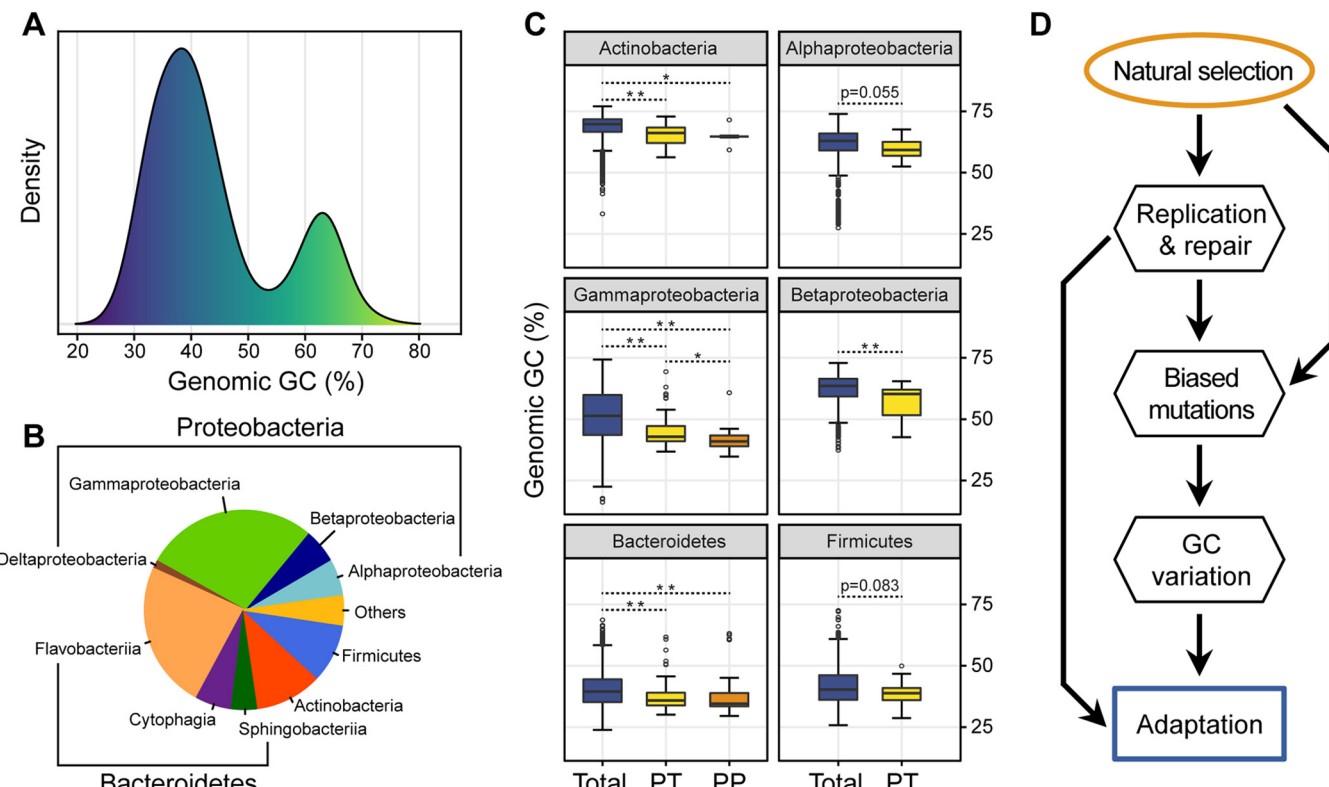

**FIG 6** Relationships between genomic GC and ancient bacterial adaptation. (A) Taxonomic composition of identified cold-adapted species. (B) Distribution of the genomic GC of psychrophilic species. (C) Comparison of the genomic GC of total, psychrotolerant (PT), and psychrophilic (PP) species within each clade. *, adjusted $P < 0.05$; **, adjusted $P < 0.01$. (D) Evolutionary mechanism of the genomic GC in bacteria.

distribution of their genomic GC, hinting at a possible link between the history of cold adaptation and the reduction of genomic GC (Fig. 6A, Fig. S15). These species mainly belong to the *Proteobacteria* (especially the *Gammaproteobacteria*) and *Bacteroidetes*, and particularly, 25% of them come from low-GC lineages in the *Gammaproteobacteria* (Fig. 4C and 6B). Such a reduction of the genomic GC can also be detected within each clade (Fig. 6C). Despite the lack of information on ancient bacterial adaptation, our results still provide a glimpse into the evolution of genomic GC content in the context of major environmental changes.

## DISCUSSION

A long-standing controversy exists regarding what mechanism accounts for the great variation in the GC content of bacterial genomes (5, 8, 46). Based on the modern synthesis and neutral theory, previous studies have mostly focused on direct selection imposed by environments and mutational biases caused by DRR-related proteins (10, 12, 13, 18, 30, 31). By performing comparative genomic analyses, we have provided more information about the variation of the genomic GC content, which potentially contributes to an improved understanding of the evolutionary mechanism. The results of our study demonstrated a phylogenetically constrained bimodal distribution of the genomic GC. This should negate a consistent rapid evolution of the genomic GC in changing environments via direct selection (15). We further showed that the evolution of the DRR system mainly accounts for the variation in the genomic GC, which again highlights the importance of mutational biases. Previous observations that even some high-GC bacteria show an AT-biased mutational pattern (17, 47) are not contradictory to this hypothesis. According to our findings, the extant values of the genomic GC of most bacteria are principally determined by their phylogenetic history in early time windows. However, this does not mean that our results support the neutral theory or neutral fixation of biased mutations. On the contrary, the genomic GC deeply influences the amino acid usage even in highly conserved regions of the

ribosomal proteins which are unlikely neutral (Fig. S16). This implies that the fixation of biased mutations still satisfies the various constraints imposed by environmental conditions, but of course, not through the resulting tiny variation of genomic GC in populations. All these findings lead to the viewpoint put forward recently that adaptation at the molecular level can be diverse and the mutational biases can shape the course of adaptive evolution in bacteria (48, 49).

Our global comparison of the functional inventory reveals that the ancestral lineages of low-GC bacteria have undergone parallel loss of various stress-related genes, especially those involved in DNA repair, suggesting effects from adaptive evolution. In previous studies, many cases regarding the combination of GC-content reduction and genome streamlining in prokaryotes have been reported (50–54). Two hypotheses, genetic drift under relaxed selection and natural selection due to nutrient restriction have been developed by researchers regarding the symbiotic and free-living bacteria, respectively (54–56). Our results indicate that a substantial selection from oligotrophic growth likely was involved in this process in most free-living bacteria. Nevertheless, given the importance of most DRR-related genes (22, 25), a relaxation in one direction of the environmental constraints on them should be necessary as well for their loss in free-living bacteria (57, 58). From a thermodynamic perspective, bacterial genomes are less susceptible to DNA damages, including oxidative damage, at low temperatures, thus in theory requiring fewer repair-related genes (59, 60). Accordingly, reduced ability of DNA repair and the loss of functional genes have been observed in some cold-adapted species (61–65). In Earth's history, major glaciation events took place multiple times from the early to later Archaean and have deeply influenced the evolution of life (42, 45). It has been reported that bacteria descended from a thermophilic ancestor and adapted to lower temperatures subsequently (66). Additionally, massive gene loss in the early evolutionary history of *Prochlorococcus*, a clade with a small genome size and fairly low GC content, has been proposed to be induced by the Snowball Earth event in a recent study (67). Therefore, the acclimation to extremely cold environments might be one of the major promoters of gene loss and GC-content divergence in bacterial genomes.

Each pathway in the DRR system recruits enzymes with diverse functions, introducing intricate effects on the genomic GC (19, 26, 68). Thus, it is difficult to speculate about the specific molecular mechanism driving the GC content divergence. Nevertheless, from an overall perspective, both previous studies and our study suggest that a streamlined DRR system (comprising the DR and postreplication MMR) leads to AT-richness (18, 69, 70). Several factors might contribute to this. First, the higher availability of A and T, possibly due to the lower energy cost of dATP and dTTP and the major role of ATP as an energy currency, likely promotes the mis-incorporation of the two bases (7). Second, the deamination of DNA cytosine residues, which is more frequently in the leading and coding strand, reduces the GC content (22). Third, the oxidation of DNA guanine residues results in mispairing of A with 8OG (oxidized guanine) during replication (25). Based on this, the repair pathways before and during DNA replication, including HR, BER, NER, and TLS, should be GC-rich to neutralize the replication bias, although no consensus has been reached. In particular, the error-prone DNA polymerases implicated in not only TLS but also HR may directly increase the genomic GC, which has been observed in laboratory (26, 28). Natural selection may break the mutational equilibrium through its influences on the DRR system and produce mutational biases.

Different adaptations have resulted in different DRR systems, which does not mean, however, that the mutational biases that changed the genomic GC in the early evolution will persist. In fact, a recent study reported that the GC-content evolution in bacteria is dominated by rare pulsed evolution instead of gradual evolution, as predicted by the punctuated equilibrium (71). A new mutational equilibrium may form, and the genomic GC will undergo long periods of stasis due to purifying selection, causing the high level of phylogenetic inertia observed in this study. This implies that the observed correlated genes in our study have not necessarily led to the corresponding changes in genomic GC directly. For example, DnaE2 might have evolved in response to the overuse of Pol V (DinB), which incorporates the oxidized dGTP opposite template A with a

high probability and increases the genomic GC (25, 27). Environmental stresses can indeed induce the overexpression of error-prone DNA polymerases to increase bacterial adaptiveness by generating adaptive mutations (72, 73). This agrees with our speculation that ancestral lineages of high-GC bacteria likely have suffered from strong oxidative and heat stresses, thus retaining more repair-related genes. These bacteria might have survived near shallow hot springs around volcanic islands during the global glaciation (74).

Overall, our study demonstrates that the legacies of ancient adaptation of bacteria still account for the major variation in genomic GC. Natural selection has driven genomic GC in an indirect way, where the adaptation of DRR induces adaptive fixation of biased mutations, indicating a new model for bacterial evolution (Fig. 6D). This idea is plausible, for example, when considering that the losses of MutM and MutT, which can lead to AT-bias and GC-bias respectively under oxidative stress (21), can both be observed in some of the low-GC clades (Fig. S9). Moreover, the genomic GC significantly influences the average amino acid properties of proteomes, including the N/C ratio and the hydrophobicity (Fig. S17), which may in turn shape bacterial fitness (6, 8). In other words, selection does not directly drive the evolution of the genomic GC, but instead acts on the striking differences of amino acid properties between high- and low-GC bacteria, thereby causing the observed correlations between genomic GC and environmental conditions in some cases (5). Our results are in agreement with many previous studies showing that the GC contents of microbiomes are under constraints from both phylogeny and the environment (10, 13, 30). The molecular mechanisms underlying the effects of the DRR system on genomic GC and, especially, their evolutionary history, of course, require further investigation. Still, our findings should provide more insights into the evolutionary mechanisms of genomic GC and promote discussion about what the microbes have brought to the modern synthesis (75).

## MATERIALS AND METHODS

**Genomic data collection and phylogenetic analysis.** A total of 11,502 NCBI-defined representative genomes were downloaded from the NCBI RefSeq database (https://www.ncbi.nlm.nih.gov/refseq/) on 30 March 2020. Of these, 11,083 and 419 genomes were classified as bacteria and archaea, respectively (Table S1). Quality measures, including the completeness and the contamination of genomes, were estimated using CheckM (76). For phylogenetic analysis, one genome with the highest quality was sampled from all genomes for each family according to their classification in the NCBI Taxonomy (https://www.ncbi.nlm.nih.gov/taxonomy/). A set of 381 protein markers ware extracted from genomes and then aligned using MAFFT v7.427 as previously described (77). All alignments were individually trimmed to remove poorly aligned positions using trimAl v1.4.15 (78). The trimmed alignments were then concatenated into a supermatrix. A maximum likelihood phylogenetic tree was constructed from the concatenated alignments using RAxML v8.2.12 under the PROTGAMMALG model, followed by 100 bootstrap replicates (79). In this tree the archaeal genomes were used as the outgroup. Similarly, the phylogenetic tree for *Gammaproteobacteria* was constructed with genomes belonging to *Betaproteobacteria* as outgroups. All trees were visualized using iTOL (80).

**Functional annotation of genomes.** All genomes were annotated with KOs using the hidden Markov models database of KEGG Orthologs (KOfam, downloaded from ftp://ftp.genome.jp/pub/db/kofam/ on 14 April 2020) implemented in HMMER v3.1b2 with the following parameters: E value, $\leq 10^{-5}$; alignment coverage, $\geq 0.7$ (81). The genome-KO matrix was then normalized to a presence/absence scale (0 for absence, 1 for presence) to exclude the effects of genome size on the conservation of gene content. The KOs, which can be assigned to KEGG pathways of DNA replication (ko03030), base excision repair (ko03410), nucleotide excision repair (ko03420), mismatch repair (ko03430), homologous recombination (ko03440), and nonhomologous end-joining (ko03450) and BRITE categories of DNA replication proteins (ko03032) and DNA repair and recombination proteins (ko03400), are defined as DRR-related KOs. Pair-wise Euclidean distance was calculated on the genome-KO matrix to indicate the intergenomic variation in genome function or DRR system. Conservation of a KO in a clade was calculated as the proportion of genomes with the presence of this KO. A genome-pathway matrix was obtained by merging the KOs in each pathway and scaling to sum to 100%.

**Molecular dating of phylogenetic trees.** Divergence times of the constructed bacterial tree using 381 protein markers were first estimated using the R package ape, with maximum constraints of 4.52 billion years ago (Ga) and 3.225 Ga for the root and the crown *Cyanobacteria*, respectively (82). Ancestral state reconstruction of the genomic GC was then carried out using the R package phytools. For a more accurate molecular clock, a reduced bacterial tree was reconstructed following the same procedures for the sake of computational expense by randomly sampling three genomes in each phylum (class in *Proteobacteria*) from the above-described bacterial tree. Moreover, three nonrepresentative genomes belonging to the newly named Melainabacteria (assembly accession no. GCA_001858525,

GCA_002102725, and GCA_013216135) were downloaded and included in this tree (Fig. S18). The divergence times were then estimated using the approximate likelihood calculation in MCMCtree v4.9 under the LG model (83). The node splitting the *Cyanobacteria* and the *Melainabacteria* was constrained to 2.5 to 2.6 Ga (77). The convergence of MCMC samples was tested in Tracer v1.7.1 (84). Furthermore, the divergence time estimation was also performed using the *Gammaproteobacteria* tree with a maximum constraint of 2.19 Ga for the root according to the result of the bacterial tree. All time-calibrated trees were visualized using FigTree v1.4.4 (https://github.com/rambaut/figtree/).

**Identification of cold-adapted species.** In order to collect possible cold-adapted bacterial species, literature searches were conducted with the following keywords: cryosphere, psychrophilic, psychrotolerant, Antarctic, arctic, glacier/glacial, and deep-sea. Only species presented in our data set were considered for the further analysis (Table S3). We also collected data from the BacDive database to obtain more information about growth temperatures (85). Here, psychrophilic bacteria are organisms growing well from below 5 to about 20℃ with an optimum no more than 15℃ (86), while the psychrotolerant bacteria are organisms that grow optimally between 15 and 25℃ and show no growth at above 30℃ (86).

**Quantification of base composition bias.** The genome size, genomic GC content, and amino acid and codon abundance in all genomes were calculated using custom Python or Perl scripts. To evaluate the concordance on the genomic GC content, coding and noncoding sequences were separately calculated as $GC_{CDS}$ and $GC_{NCS}$, respectively. Furthermore, the GC content contributed by amino acid usage was calculated by fixing the synonymous codon usage:

$$GC_{AA} = \sum_{AA} A_i \cdot AC_i$$

where $A_i$ is the abundance of amino acids (AA) in a genome and $AC_i$ is the average codon GC content of an amino acid provided that there is no bias in synonymous codon usage. The GC content contributed by synonymous codon usage was calculated by fixing the amino acid usage:

$$GC_{Codon} = \sum_{AA} \sum_{SC} AA_i \cdot R_j \cdot C_j$$

where $AA_i$ is the average abundance of amino acids in all genomes, $R_j$ is the relative abundance of a synonymous codon (SC) of an amino acid, and $C_j$ is the GC content of this codon.

**Quantification of amino acid properties.** To evaluate the influence of variation in amino acid composition, chemical properties, including the molecular weight, VdW (Van der Waals) volume, isoelectric point (pI), hydrophobicity, polarity, number of nitrogen atoms (N number), and N/C ratio, were collected via literature searches (Table S4). The average properties of each bacterial proteome were calculated based on the amino acid composition. Taking average hydrophobicity as an example,

$$\text{Average hydrophobicity} = \sum_{AA} A_i H_i$$

where $H_i$ is the hydrophobicity of a specific amino acid.

**Phylogenetic comparative methods.** Two measures of phylogenetic signal, Blomberg's $K$ and Pagel's $\lambda$, were estimated on the average genomic GC of each bacterial family in the constructed bacterial tree using the phytools package in R (87). A high value of Blomberg's $K$ (0 to infinity) or Pagel's $\lambda$ (0 to 1) indicates a strong phylogenetic signal (87). The kappa ($\kappa$) and delta ($\delta$) were also estimated using the Geiger package. A high value of $K$ represents punctuated evolution, and a high value of $\delta$ (>1) represents accelerated evolution (88). Moreover, phylogenetically independent contrasts were computed on the genomic GC data and the genome-pathway matrix to test the variation pattern of DRR without phylogenetic influence using the ape package. A phylogenetic generalized least-squares (PGLS) regression of the data matrix comprising conservation data of DRR-related KOs in each bacterial family was performed to explain the average genomic GC using the caper package (88).

**Other statistical analyses and visualization.** Statistical analysis, including the calculation of Pearson's correlation coefficient and multiple regression analysis, and visualization were mainly performed using R v4.0.3. Variance of the genomic GC unexplained by taxonomy is calculated based on the sum of squared deviations (SS):

$$\text{Variance unexplained} = 1 - \frac{SS_w}{SS_T}$$

where $SS_T$ is the total SS of the genomic GC and $SS_W$ is the sum of within-taxon SS at a specific taxonomic level. Wilcoxon tests were conducted to compare the intergroup differences in genomic GC. All estimated $P$ values in multiple comparisons were adjusted using the Bonferroni method. The 16 highly conserved ribosomal proteins in bacteria were extracted from the proteins of each genome as described in a previous study (89) and then aligned and trimmed equally as described above. Finally, the variance partitioning analyses (VPA) of the amino acid composition of whole-genome proteins and ribosomal protein alignments were performed using the vegan package (https://cran.r-project.org/web/packages/vegan/index.html).

## SUPPLEMENTAL MATERIAL

Supplemental material is available online only.
**SUPPLEMENTAL FILE 1**, PDF file, 2.1 MB.
**SUPPLEMENTAL FILE 2**, XLSX file, 0.4 MB.
**SUPPLEMENTAL FILE 3**, XLSX file, 0.03 MB.
**SUPPLEMENTAL FILE 4**, XLSX file, 0.01 MB.

## ACKNOWLEDGMENTS

We thank Guisong Chen from Guangdong Magigene Biotechnology Co., Ltd., for assistance with the writing of Python scripts for the data analyses.

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
