## [Reviewer comments · Microbiology Spectrum]

Microbiology Spectrum

Genomic legacies of ancient adaptation illuminate GC-content evolution in bacteria

Wenkai Teng, Bin Liao, Mengyun Chen, and Wensheng Shu

Corresponding Author(s): Wensheng Shu and Mengyun Chen, Institute of Ecological Science, School of Life Science, South China Normal University

Review Timeline:

Submission Date:	June 7, 2022
Editorial Decision:	September 13, 2022
Revision Received:	October 22, 2022
Accepted:	December 1, 2022

Editor: Sébastien Faucher

Reviewer(s): The reviewers have opted to remain anonymous.

Transaction Report:

DOI: <https://doi.org/10.1128/spectrum.02145-22>

September 13, 2022

Prof. Wensheng Shu
Institute of Ecological Science, School of Life Science, South China Normal University
Guangzhou
China

Re: Spectrum02145-22 (Genomic legacies of ancient adaptation illuminate the GC-content evolution in bacteria)

Dear Prof. Wensheng Shu:

Your manuscript was reviewed by two experts in this field of research. Both found the results presented interesting but raised a few concerns that needs to be address. The method and statistical analysis needs to be described in further details. One of the reviewer also suggested a modification to the analysis that should strengthen the conclusions. Both reviewers noted that the languages needs improvement.

Link Not Available

Sincerely,

Sébastien Faucher

Journals Department
Reviewer comments:

Reviewer #1 (Comments for the Author):

Review

A rigorous study of GC-content evolution in bacteria emphasizing the indirect effects on natural selection. Excellent figures and text that only requires minor linguistic modifications (see attached edited file and specific suggestions below).

I appreciated that the paper follows recent understanding of the impact of mutational biases imposed by DRR on genome evolution and that these are "indirect by-products of processes operating at lower levels of organization" in the words of Lynch

(2018). The authors make the case that a more comprehensive view is needed in understanding the evolution of life by looking at both direct and indirect consequences of natural selection and propose a model where DRR is influencing both biased mutation and adaptation (Fig. 3D) which I endorse.

Missing a speculation as to why DRRs positively (BER, TLS, NHEJ and NER) versus negatively (DR, HR and MMR) correlate with genomic GC (directly, indirectly or both).

Missing a consideration of how stress-induced mutagenesis by means of DRR-related proteins described can contribute to adaptation in its own right (e.g. reviewed by Susan M. Rosenberg).

Define KOs

L35 Suggest "as well as being environment associating" or equivalent phrasing

L82 Change to "contribute"

L95 Suggest change "collected" to "analysed"

L107 Suggest: "have GC contents either below..."

L114 Suggest "values either greater..."

L117 Change to "The results above"

L143 Suggest "that challenge" or "challenging"

L214 Change to "correlating"

L296 Change to "growth likely were involved"

L325 Change to "are in agreement that"

L391 Change "now" to "no"

L429 "were extracted"

Reviewer #2 (Comments for the Author):

The drivers of prokaryotic genomic GC content variation have long been a puzzle often separated into two categories: mutation bias and fixation bias. In this paper, Teng et al. primarily address the former category. They analyze ~11k complete RefSeq prokaryotic genomes to suggest that GC variation stems primarily from corresponding variation in DNA replication and repair (DRR) processes across the prokaryotic tree of life. They next propose that the DRR pathways, in turn, evolve due to selection arising from shifts in environmental conditions. Their arguments are plausible, although a key model needs to be modified in order to make their argument compelling. Further, the methods section needs greater detail in order to understand exactly how their analyses were conducted.

1. The central conclusion of this paper is that changes to DRR deep in the phylogeny explain GC content. However, the statistical model used to explain GC using DRR genes as predictors appears to be a simple linear model that treats all genomes as independent, which is not a reasonable assumption. This model needs to be replaced with one that explicitly accounts for phylogeny and looks at the effect of DRR only after taking into account the phylogenetic correlation structure - otherwise the observed correlation may simply be DRR serving as proxy for phylogeny. Phylogenetic generalized least squares (PGLS) analysis would be a reasonable approach. Further, the current methods section does not explain how exactly DRR genes are used as predictors (e.g., are these whole-gene binary presence / absence? specific amino acid changes?).

2. In general, the results in Fig 3 (the key connection between GC and DRR) are hard to interpret as-is. They need greater explanation and to include measures of uncertainty. Part B does not define the "minimum distance in DRR" - is this some sort of edit distance? How are distances combined across genes? Part C shows genes with high correlations, but what is being correlated and how much data underlies each correlation value? These are the top genes, but how many genes were examined in total? There is no measure or way for the reader to assess statistical significance.

3. Fig 4 links DRR genes with environmental factors, but how were these particular genes chosen to be presented? Was the analysis performed on all genes and only the top correlations reported? Or were genes chosen to be functionally interesting?

4. The selection arguments are grounded in associations between DRR variation and environmental conditions that initially appear more anecdotal than quantitative (i.e., Fig 4A and lines 232-249). The results from cold-adapted species are more compelling, although that is only a single environmental condition.

Specific points

- define KO upon first use (at first, I thought it meant "knockout" but I later realized you reference "KEGG ortholog"..)
- line 107: meaningless statement: "most genomes have GC contents below or above 50%"
- line 122: how define "group" in context of "same bacterial group"
- line 145: I think you wanted a different preposition than "other" but I'm not sure what you meant.

- line 145: more clear if made the list parallel including the first item: "Secondly, the GC content genome wide (GC), the GC content of coding sequences (GCcds), the GC ..."
- line 149-150: sentence is confusing
- line 160: which previous scenarios? This sounds like it needs a citation.
- fig 4B: how define "minimum distance in function"?
- fig 4D: colors inside matrix need a key
- line 214: typo: "severer"
- line 364: what genes were used for molecular dating of the phylogenetic trees?
- line 492: typo: "ware"

Staff Comments:

Preparing Revision Guidelines

Please return the manuscript within 60 days; if you cannot complete the modification within this time period, please contact me. If you do not wish to modify the manuscript and prefer to submit it to another journal, please notify me of your decision immediately so that the manuscript may be formally withdrawn from consideration by Microbiology Spectrum.

Response to Reviewers' Comments

We greatly appreciate the editor and reviewers for their constructive comments and suggestions to improve the quality of the paper. Those comments are all valuable and very helpful for revising and improving our paper. We have studied comments carefully and have made corrections which we hope to meet with approval. We present a point-by-point response to the reviewers' comments in the following part (the replies are highlighted in blue).

Reviewer #1 (Comments for the Author):

Review

A rigorous study of GC-content evolution in bacteria emphasizing the indirect effects on natural selection. Excellent figures and text that only requires minor linguistic modifications (see attached edited file and specific suggestions below).

I appreciated that the paper follows recent understanding of the impact of mutational biases imposed by DRR on genome evolution and that these are "indirect by-products of processes operating at lower levels of organization" in the words of Lynch (2018). The authors make the case that a more comprehensive view is needed in understanding the evolution of life by looking at both direct and indirect consequences of natural selection and propose a model where DRR is influencing both biased mutation and adaptation (Fig. 3D) which I endorse.

Missing a speculation as to why DRRs positively (BER, TLS, NHEJ and NER) versus negatively (DR, HR and MMR) correlate with genomic GC (directly, indirectly or both).

We have added the discussion in the revised manuscript:

(L. 329-346) "Each pathway in the DRR system recruit enzymes with diverse functions introducing intricate effects on the genomic GC (19, 26, 68). Thus, it is difficult to speculate the specific molecular mechanism driving the GC content divergence. Nevertheless, from an overall perspective both previous studies and our study suggest that streamlined DRR system (comprising the DR and post-replication MMR) leads to AT-rich (18, 69, 70). Several factors might contribute to this. First, higher availability of A and T, possibly due to the lower energy cost of dATP and dTTP and the major role of ATP as an energy currency, likely promote the mis-incorporation of the two bases (7). Second, the deamination of DNA cytosine residues, which is more frequently in the leading and coding strand, will reduce the GC content (22). Third, the oxidation of DNA guanine residues will

result in mispairing of A with 8OG (oxidized guanine) during replication (25). According to this the repair pathways before and during DNA replication, including the HR, BER, NER and TLS, should be GC-rich to neutralize the replication bias although no consensus has been reached. Particularly, the error-prone DNA polymerases implicated in not only TLS but also HR may directly increase the genomic GC which has been observed in laboratory (26, 28). Natural selection may break the mutational equilibrium through its influences on the DRR system and produce mutational biases."

(L. 348-358) "Different adaptations have resulted in different DRR systems, which does not mean, however, the mutational biases having changed the genomic GC in the early evolution persist. In fact, as a recent study reported the GC-content evolution in bacteria is dominated by rare pulsed evolution instead of gradual evolution, as predicted by the punctuated equilibrium (71). New mutational equilibrium may form and the genomic GC will undergo long periods of stasis due to purifying selection, causing a strong level of phylogenetic inertia observed in this study. This implies that the observed correlated genes in our study have not necessarily led to the corresponding changes in the genomic GC directly. For example, the DnaE2 might have evolved in response to the overuse of Pol V (DinB) which incorporates the oxidized dGTP opposite template A with a high probability increasing the genomic GC (25, 27)."

Missing a consideration of how stress-induced mutagenesis by means of DRR-related proteins described can contribute to adaptation in its own right (e.g. reviewed by Susan M. Rosenberg).

We have added the discussion in the revised manuscript:

(L. 358-364) "Environmental stresses can indeed induce the overexpression of error-prone DNA polymerases to increase the bacterial adaptiveness by generating adaptive mutations (72, 73). This agrees with our speculation that ancestral lineages of high-GC bacteria likely have suffered from strong oxidative and heat stresses, thus retaining more repair-related genes. These bacteria might have survived near shallow hot springs around volcanic islands during the global glaciation (74)."

Define KOs

We have defined KOs at first use in the revised manuscript:

(L. 160-163) "As anticipated, a linear model using 217 DRR-related KOs (KEGG orthologs) can explain up to 88% of the total variance (that is, with a multiple

correlation coefficient of 0.94) even taking phylogenetic relationships into account, suggesting that the DRR system can serve as a good predictor of the genomic GC (Fig. 3A)."

L35 Suggest "as well as being environment associating" or equivalent phrasing

We have rewritten the phrasing in the revised manuscript as suggested:

(L. 35-36) "..., high intragenomic homogeneity, strong level of phylogenetic inertia, as well as being environment associating."

L82 Change to "contribute"

We have corrected it in the revised manuscript as suggested:

(L. 81-83) "The proteins involved in these pathways are critical for the response to environmental stresses and deletions of them could contribute to an elevated rate of related mutations (25)."

L95 Suggest change "collected" to "analysed"

We have corrected it in the revised manuscript as suggested:

(L. 94-95) "In order to clarify the evolutionary mechanisms of the genomic GC, we analyzed all currently available representative genomes of bacteria. "

L107 Suggest: "have GC contents either below..."

We have corrected it in the revised manuscript as suggested:

(L. 106-108) "This indicates a remarkable divergence of the genomic GC as most genomes have GC contents either below 45% or above 60% (Fig. 1A)."

L114 Suggest "values either greater..."

We have corrected it in the revised manuscript as suggested:

(L. 111-113) "Distributions of the genomic GC in most phyla are approximatively unimodal with peak values either greater or less than 50%..."

L117 Change to "The results above"

We have corrected it in the revised manuscript as suggested:

(L. 119-120) "The results above suggest that early evolution of bacteria contributes much to the divergence of the genomic GC, which leads to the bimodal distribution."

L143 Suggest "that challenge" or "challenging"

We have corrected it in the revised manuscript as suggested:

(L. 146-147) "However, there are several findings challenging this view."

L214 Change to "correlating"

We have corrected it in the revised manuscript as suggested:

(L. 227-229) "Second, there are much more KOs positively correlating with the genomic GC, which implies more severe loss of genes in low-GC clades (Fig. S12)."

L296 Change to "growth likely were involved"

We have corrected it in the revised manuscript as suggested:

(L. 310-311) "Our results indicate that a substantial selection from oligotrophic growth likely was involved in this process in most free-living bacteria."

L325 Change to "are in agreement that"

We have rewritten it in the revised manuscript as suggested:

(L. 378-380) "Our results are in agreement with many previous studies that the GC contents of microbiomes are under constraints from both phylogeny and environments (10, 13, 30)."

L391 Change "now" to "no"

We have corrected it in the revised manuscript as suggested:

(L. 448-449) "While the psychrotolerant bacteria are organisms that grow optimally between 15 and 25 °C and show no growth at above 30 °C (86)."

L429 "were extracted"

We have corrected it in the revised manuscript as suggested:

(L. 500-502) " The 16 highly conserved ribosomal proteins in bacteria were extracted from the proteins of each genome as described in previous study (89) and then aligned and trimmed equally as described above."

Reviewer #2 (Comments for the Author):

The drivers of prokaryotic genomic GC content variation have long been a puzzle often separated into two categories: mutation bias and fixation bias. In this paper, Teng et al. primarily address the former category. They analyze ~11k complete RefSeq prokaryotic

genomes to suggest that GC variation stems primarily from corresponding variation in DNA replication and repair (DRR) processes across the prokaryotic tree of life. They next propose that the DRR pathways, in turn, evolve due to selection arising from shifts in environmental conditions. Their arguments are plausible, although a key model needs to be modified in order to make their argument compelling. Further, the methods section needs greater detail in order to understand exactly how their analyses were conducted.

1. The central conclusion of this paper is that changes to DRR deep in the phylogeny explain GC content. However, the statistical model used to explain GC using DRR genes as predictors appears to be a simple linear model that treats all genomes as independent, which is not a reasonable assumption. This model needs to be replaced with one that explicitly accounts for phylogeny and looks at the effect of DRR only after taking into account the phylogenetic correlation structure - otherwise the observed correlation may simply be DRR serving as proxy for phylogeny. Phylogenetic generalized least squares (PGLS) analysis would be a reasonable approach. Further, the current methods section does not explain how exactly DRR genes are used as predictors (e.g., are these whole-gene binary presence / absence? specific amino acid changes?).

We have improved our analyses in the revised manuscript as suggested. Firstly, we estimated the Blomberg's K, Pagel's λ , kappa and delta to quantitatively show the features of GC-content evolution:

(L. 115-119) "Strong phylogenetic signal can also be reflected by the high values of Blomberg's K ($K = 1.47$) and Pagel's λ ($\lambda = 0.998$) in the bacterial tree (Fig. 2). The estimated kappa ($k = 0.842$) and delta ($\delta = 0.567$) on genomic GC further suggest a punctuated and decelerated evolution mode (Fig. 2, see Methods)."

Secondly, phylogenetic generalized least squares (PGLS) analysis was applied to fit the genomic GC using DRR-related KOs based on the bacterial tree (Fig. 2):

(L. 160-163) "As anticipated, a linear model using 217 DRR-related KOs (KEGG orthologs) can explain up to 88% of the total variance (that is, with a multiple correlation coefficient of 0.94) even taking phylogenetic relationships into account, suggesting that the DRR system can serve as a good predictor of the genomic GC (Fig. 3A)."

Thirdly, we tested the evolutionary pattern of DRR pathways in a phylogenetic independent way:

(L. 191-196) " This variation pattern could also be observed when excluding the phylogenetic influence by performing phylogenetically independent contrast, with

one exception (the insignificant negative correlated NER) (Fig. S6). In detail, consistent with the inferred parallel divergences of genomic GC, a similar evolutionary pattern of the DRR exists in all three major clades (Fig. 2, Fig. S7)."

Details of the methods have been added in the revised manuscript:

(L. 478-489) "Two measures of phylogenetic signal, the Blomberg's K and Pagel's λ , were estimated on the average genomic GC of each bacterial family in the constructed bacterial tree using the phytools package in R (87). A high value of the Blomberg's K (0-infinity) or Pagel's λ (0-1) indicates a strong phylogenetic signal (87). The kappa (k) and delta (δ) were also estimated using the Geiger package. A high value of k represents punctuated evolution and A high value of δ (> 1) represents accelerated evolution (88). Moreover, phylogenetically independent contrasts were computed on the genomic GC data and the genome-pathway matrix to test the variation pattern of DRR without phylogenetic influence using the ape package. A phylogenetic generalized least squares (PGLS) regression of the data matrix comprising conservation data of DRR-related KOs in each bacterial family was performed to explain the average genomic GC using the caper package (88)."

2. In general, the results in Fig 3 (the key connection between GC and DRR) are hard to interpret as-is. They need greater explanation and to include measures of uncertainty. Part B does not define the "minimum distance in DRR" - is this some sort of edit distance? How are distances combined across genes? Part C shows genes with high correlations, but what is being correlated and how much data underlies each correlation value? These are the top genes, but how many genes were examined in total? There is no measure or way for the reader to assess statistical significance.

We have improved our explanation in the revised manuscript as suggested.

For Fig. 3B:

(L. 163-169) "In order to test whether a change in DRR system is necessary to alter the genomic GC, we analyzed the GC-content difference and DRR distance (a Euclidean distance calculated based on the presence of DRR-related KOs) among all genome pairs. Good agreement was achieved between the GC-content difference and allowed minimum DRR distance (the minimum distance in DRR of genome pairs within 1% range of GC-content difference, e.g., 0-1%, 1-2%...), providing further support for our hypothesis (Fig. 3B)."

For Fig. 3C:

(L. 170-171) "Correlation coefficients were also quantified for detecting the relationships between GC content and 231 DRR-related KOs detected in total using

11,083 genomes."

(L. 827-829) "(C) Correlations between the genomic GC and DRR-related KOs. Of all 231 KOs, only the one with a considerable (Pearson's $r > 0.4$ or $r < -0.4$) correlation is shown."

Details of the methods have been added in the revised manuscript:

(L. 409-420) "The genome-KO matrix was then normalized to presence/absence scale (0 for absence, 1 for presence) to exclude the effects of genome size on conservation of gene content. The KOs which can be assigned to KEGG pathways of DNA replication (ko03030), base excision repair (ko03410), nucleotide excision repair (ko03420), mismatch repair (ko03430), homologous recombination (ko03440), and non-homologous end-joining (ko03450) and BRITE categories of DNA replication proteins (ko03032) and DNA repair and recombination proteins (ko03400) are defined as DRR-related KOs. Pair-wise Euclidean distance was calculated on the genome-KO matrix to indicate the intergenomic variation in genome function or DRR system. Conservation of a KO in a clade was calculated as the proportion of genomes with the presence of it. A genome-pathway matrix was obtained by merging the KOs in each pathway and scaling to sum to 100%."

3. *Fig 4 links DRR genes with environmental factors, but how were these particular genes chosen to be presented? Was the analysis performed on all genes and only the top correlations reported? Or were genes chosen to be functionally interesting?*

In Fig 4A, all annotated KOs (9,924 in total) were analyzed and the top correlated KOs are presented:

(L. 199-120) "In order to better understand the evolutionary scenarios, a genome-wide investigation in functional variation was conducted using all annotated KOs (9,924 in total)."

(L. 840-844) "(A) Highly correlated KOs and their involvement in the bacterial response to changes in four environmental factors. Of all 9,924 KOs, only the one with a considerable (Pearson's $r > 0.6$ or $r < -0.5$) correlation are shown. Asterisks beside the bars indicate adjusted p-value < 0.01 ."

4. *The selection arguments are grounded in associations between DRR variation and environmental conditions that initially appear more anecdotal than quantitative (i.e., Fig 4A and lines 232-249). The results from cold-adapted species are more compelling, although that is only a single environmental condition.*

Thanks a lot for the remarks. We have studied the function of highly correlated DRR and none-DRR genes, both in favor of the associations between bacterial

evolution and environmental stresses especially heat and oxidative stresses:

(L. 182-187) "Many of these proteins have been shown to be crucial for growth under environmental stresses. For instance, both the MutM and Ku play important roles in combating heat as well as oxidative stresses (25, 30). Beyond that, the MutT and its analogue MutT1 participate in the detoxification of oxidized guanine nucleotides (21, 36). TLS polymerases are also important for growth under harsh environments such as starvation and high temperatures (26)."

(L. 202-214) "Similar to the situation in DRR system, the highly correlated proteins can be related to the bacterial response to environmental changes of multiple factors comprising oxygen, temperature, nutrition and osmotic pressure (Fig. 4A, Table S2). For instance, the YbbN, a molecular chaperone of the β -clamp in the replication complex, can promote the bacterial acclimation to higher temperatures and are mainly conserved in high-GC clades (Fig. S8). Another positively correlated protein named OtsA involves in the synthesis of trehalose in response to heat, cold and osmotic stresses (Fig. 4A, Table S2). Besides that, the FNR, SURF1 and YgfZ which play important roles in aerobic growth also positively correlate with the genomic GC. By contrast, several proteins including the PepT, ComEB and PfkA which are required for the utilization of diverse nutrient sources and anaerobic regulatory proteins FLP and Rny negatively correlate with the genomic GC."

We have added more discussion and citations as to support our hypothesis:

(L. 352-360) "Environmental stresses can indeed induce the overexpression of error-prone DNA polymerases to increase the bacterial adaptiveness by generating adaptive mutations (72, 73). This agrees with our speculation that ancestral lineages of high-GC bacteria likely have suffered from strong oxidative and heat stresses, thus retaining more repair-related genes. These bacteria might have survived near shallow hot springs around volcanic islands during the global glaciation (74)."

Specific points

- *define KO upon first use (at first, I thought it meant "knockout" but I later realized you reference "KEGG ortholog"..)*

We have defined KOs at first use in the revised manuscript:

(L. 160-163) "As anticipated, a linear model using 217 DRR-related KOs (KEGG orthologs) can explain up to 88% of the total variance (that is, with a multiple correlation coefficient of 0.94) even taking phylogenetic relationships into account,

suggesting that the DRR system can serve as a good predictor of the genomic GC (Fig. 3A)."

- *line 107: meaningless statement: "most genomes have GC contents below or above 50%"*

We have rewritten the phrasing in the revised manuscript:

(L. 106-108) "This indicates a remarkable divergence of the genomic GC as most genomes have GC contents either below 45% or above 60% (Fig. 1A)."

- *line 122: how define "group" in context of "same bacterial group"*

We have defined it in the revised manuscript as suggested:

(L. 123-126) "Low-GC clades are actually interspersed among high-GC clades on the phylogenetic tree even within the same bacterial group (e.g., the Terrabacteria group, the Proteobacteria group, and the FCB and PVC group classified by the NCBI taxonomy: <https://www.ncbi.nlm.nih.gov/Taxonomy/>) (Fig. 2)."

- *line 145: I think you wanted a different preposition than "other" but I'm not sure what you meant.*

We have corrected it in the revised manuscript as suggested:

(L. 147-149) "Firstly, the correlation between the genomic GC and the usage of an amino acid depends on the average GC content of its codons rather than the chemical properties generally targeted by selection (Fig. S4)."

- *line 145: more clear if made the list parallel including the first item: "Secondly, the GC content genome wide (GC), the GC content of coding sequences (GC_{cds}), the GC ..."*

We have corrected it in the revised manuscript as suggested:

(L. 150-153) "Secondly, the GC content genome wide (GC), the GC content of coding sequences (GC_{CDS}), the GC content of non-coding sequences (GC_{NCS}), and the GC content contributed by the usage of amino acids (GC_{AA}) and codons (GC_{Codon}) are all highly correlated with each other (Fig. S5)."

- *line 149-150: sentence is confusing*

We have corrected it in the revised manuscript as suggested:

(L. 153-155) "Within this context, assuming an environmental factor which equivalently influences the coding and non-coding sequences via direct selection can be confusing (5)."

- *line 160: which previous scenarios? This sounds like it needs a citation.*

We have added citations in the revised manuscript as suggested:

(L. 171-173) "Unlike previous scenarios (18, 30, 31), we find that multiple DRR-related KOs are implicated in the explanation of the variation of the genomic GC (Fig. 3C)."

- *fig 4B: how define "minimum distance in function"?*

It is similar to the minimum DRR distance. We have defined it in the revised manuscript as suggested:

(L. 163-169) "In order to test whether a change in DRR system is necessary to alter the genomic GC, we analyzed the GC-content difference and DRR distance (a Euclidean distance calculated based on the presence of DRR-related KOs) among all genome pairs. Good agreement was achieved between the GC-content difference and allowed minimum DRR distance (the minimum distance in DRR of genome pairs within 1% range of GC-content difference, e.g., 0-1%, 1-2%...), providing further support for our hypothesis (Fig. 3B)."

(L. 844-846) "(B) Minimum Euclidean distance in genome function (calculated using 9,924 KOs) with changing differences in the GC-content among all genome pairs."

- *fig 4D: colors inside matrix need a key*

We have added the key in Fig. 4D in the revised manuscript as suggested.

- *line 214: typo: "severer"*

We have corrected it in the revised manuscript as suggested:

(L. 227-229) "Second, there are much more KOs positively correlating with the genomic GC, which implies more severe loss of genes in low-GC clades (Fig. S12)."

- *line 364: what genes were used for molecular dating of the phylogenetic trees?*

We have indicated the markers used for phylogenetic analyses and molecular dating in the revised manuscript:

(L. 395-396) "A set of 381 protein markers were extracted from genomes and then aligned using MAFFT v7.427 as previously described (77)."

(L. 423-425) "Divergence times of the constructed bacterial tree using 381 protein markers were firstly estimated using the R package APE, with maximum

constraints of 4.52 Ga and 3.225 Ga for the root and the crown Cyanobacteria, respectively (82)."

- *line 492: typo: "ware"*

We have corrected it in the revised manuscript as suggested:

(L. 500-502) "The 16 highly conserved ribosomal proteins in bacteria were extracted from the proteins of each genome as described in previous study (89) and then aligned and trimmed equally as described above."

December 1, 2022

Prof. Wensheng Shu
Institute of Ecological Science, School of Life Science, South China Normal University
Guangzhou
China

Re: Spectrum02145-22R1 (Genomic legacies of ancient adaptation illuminate GC-content evolution in bacteria)

Dear Prof. Wensheng Shu:

Your manuscript has been accepted, and I am forwarding it to the ASM Journals Department for publication. You will be notified when your proofs are ready to be viewed.

Sincerely,

Sébastien Faucher
Editor, Microbiology Spectrum

Journals Department
Supplemental Material: Accept
Supplemental Material: Accept
Supplemental Material: Accept
Supplemental Material: Accept